# Treatment with Ad5-Porcine Interferon-α Attenuates Ebolavirus Disease in Pigs

**DOI:** 10.3390/pathogens11040449

**Published:** 2022-04-08

**Authors:** Chandrika Senthilkumaran, Andrea L. Kroeker, Gregory Smith, Carissa Embury-Hyatt, Brad Collignon, Elizabeth Ramirez-Medina, Paul A. Azzinaro, Bradley S. Pickering, Fayna Diaz-San Segundo, Hana M. Weingartl, Teresa de los Santos

**Affiliations:** 1National Centre for Foreign Animal Disease (NCFAD), Canadian Food Inspection Agency, Winnipeg, MB R3E 3M4, Canada; senthil.chandrika@gmail.com (C.S.); andrea.kroeker@usask.ca (A.L.K.); greg.smith2@canada.ca (G.S.); carissa.emburyhyatt@canada.ca (C.E.-H.); bcollignon@sharedhealthmb.ca (B.C.); bradley.pickering@canada.ca (B.S.P.); hanamarieweingartl@seznam.cz (H.M.W.); 2Public Health Ontario, Santé Publique Ontario, Toronto, ON M5G 1M1, Canada; 3VIDO-InterVac, University of Saskatchewan, Saskatoon, SK S7N 5E3, Canada; 4Plum Island Animal Disease Center, North-East Area, Agricultural Research Service, U.S. Department of Agriculture, Orient, NY 11957, USA; elizabeth.ramirez@usda.gov (E.R.-M.); paul.azzinaro@usda.gov (P.A.A.); fayna.diaz-sansegundo@usda.gov (F.D.-S.S.); 5Department of Medical Microbiology, Faculty of Medicine, University of Manitoba, Winnipeg, MB R3E OW2, Canada

**Keywords:** ebola virus, swine, adenovirus, interferon alpha, interferon lambda, IFN

## Abstract

Under experimental conditions, pigs infected with Ebola Virus (EBOV) develop disease and can readily transmit the virus to non-human primates or pigs. In the event of accidental or intentional EBOV infection of domestic pigs, complex and time-consuming safe depopulation and carcass disposal are expected. Delaying or preventing transmission through a reduction in viral shedding is an absolute necessity to limit the spread of the virus. In this study, we tested whether porcine interferon-α or λ3 (porIFNα or porIFNλ3) delivered by a replication-defective human type 5 adenovirus vector (Ad5-porIFNα or Ad5-porIFNλ3) could limit EBOV replication and shedding in domestic pigs. Our results show that pigs pre-treated with Ad5-porIFNα did not develop measurable clinical signs, did not shed virus RNA, and displayed strongly reduced viral RNA load in tissues. A microarray analysis of peripheral blood mononuclear cells indicated that Ad5-porIFNα treatment led to clear upregulation in immune and inflammatory responses probably involved in protection against disease. Our results indicate that administration of Ad5-porIFNα can potentially be used to limit the spread of EBOV in pigs.

## 1. Introduction

The *Ebolavirus* genus belongs to the single-stranded, non-segmented negative-sense RNA virus family Filoviridae, and includes six species: *Zaire ebolavirus* (EBOV), *Sudan ebolavirus* (SUDV), *Bundibugyo ebolavirus* (BDBV), *Tai Forest ebolavirus* (TAFV), *Reston ebolavirus* (RESTV), and *Bombali ebolavirus* (BOMV) [1]. Of these species, the first four are known to cause Ebola disease (EBOD) in humans and non-human-primates; RESTV is asymptomatic in people but may cause lethal EBOD in non-human primates [2]. The first EBOD human outbreaks were described in Sudan and Zaire in 1976, and EBOV, SUDV, and BDBV have since continued to cause regular outbreaks in rural areas of Africa and have been limited to relatively small areas, a low number of people, and short periods of time. However, in 2014–2016, a major outbreak of unprecedented scale took place in Western Africa, spreading through cities and multiple countries and killing over 11,000 people [3]. EBOD outbreaks in human populations are initiated through zoonotic events, although, thus far, it has not been clearly defined which are the natural animal hosts for Ebola viruses [4]. Pigs were first implicated as an ebolavirus host in 2008 in the Philippines when RESTV RNA was identified in animals with porcine respiratory and reproductive syndrome [5]. Subsequent detection of RESTV antibodies in several workers at the affected pig farms strongly suggests zoonotic transmission of RESTV [6,7]. Although the involvement of wild or domestic pigs in the transmission cycle of African ebolaviruses has not been clearly defined, there are indications that pigs can be infected with ebolaviruses other than RESTV [8,9]. In humans, EBOD presents as a severe hemorrhagic fever with strong liver tropism, high viremia, and high mortality. In contrast, EBOD in experimentally infected pigs was mainly limited to the respiratory tract, where it caused severe but transient inflammation with relatively non-specific clinical signs [10]. The severity of disease appeared to differ based on the age of inoculated pigs, and in some instances, infection with African ebolaviruses was asymptomatic in this species. Moreover, shedding was not correlated with the severity of EBOD, but importantly, the less affected group transmitted EBOV to non-human primates without direct contact [11]. The relatively mild clinical signs of this EBOD in pigs can make it difficult to distinguish it from other swine respiratory diseases and can complicate or delay detection of EBOV-infected animals in the field. Farmers have typically attempted to control and stamp out outbreaks of infectious diseases by depopulating entire farms in combination with ring vaccination (and subsequent depopulation) to contain the outbreak. Outbreaks of EBOV in swine would have additional complexities to consider due to its zoonotic potential, with high infectivity and high mortality for people. Safe massive slaughter and carcass disposal would be logistically very challenging, and additional tools to gain more time for control would be absolutely necessary. Although several antiviral therapies and a vaccine have now been approved by the FDA for human use against EBOV [12], controlling the virus in swine populations would still be required to hamper the virus’ spread in the field. Interestingly, in safety studies, inoculation of pigs with VSV-EBOV, a live vaccine approved for humans, did not cause disease in the animals, and virus shedding was minimal [13]. If considered one of the control tools, efficacy in pigs would have to be determined, including the onset of immunity. As time would be of the essence, preventing shedding as early as possible (prior to vaccine elicited immunity) would be highly desirable. One logical strategy to consider would be the use of biotherapeutics, such as interferon molecules (IFN), that would rapidly block virus replication. In fact, this strategy has been successfully applied to rapidly control other viruses, including foot-and-mouth disease virus (FMDV), a pathogen of significant importance in the livestock industry. In swine, the administration of a replication defective human adenovirus type 5 (Ad5-Blue) expressing IFNα alone or in combination with a similar vector expressing a subunit vaccine (Ad5-FMD) elicited immediate protection prior to challenge with fully virulent FMDV [14]. Moreover, the same vector delayed and reduced the severity of other pig viral diseases, such as classical swine fever virus (CSFV) [15]. Interestingly, protection against FMDV was also obtained by the administration of type III IFN (IFNλ3) using the same Ad5-blue vector [16]. It has been shown that type I IFN might override EBOV anti-IFN functions. For instance, early studies showed that treatment with IFN-α2b reduced clinical signs and extended the time to death in a cynomolgus model of lethal EBOD [17]. Notably, although a limited number of patients were evaluated, administration of IFN-β improved the survival of EBOV-infected patients [18]. All these studies highlight the potential use of IFN to limit EBOD. In the current study, we evaluated the efficacy of porIFNα or porIFNλ3 delivered by Ad5 vectors as a prophylactic treatment against EBOV. Our results demonstrate that while type III IFN (IFNλ3) is not effective, treatment with type I IFN (IFNα) clearly prevents or reduces the severity of EBOD in pigs.

## 2. Results

### 2.1. Outcome of EBOV Inoculation of PBS-Treated Control Animals Was Consistent with Previous Laboratory Studies

We previously demonstrated that EBOV can cause disease in experimentally infected pigs [10]. In order to verify these results during the current experiment, we first analyzed the outcome of three animals subcutaneously (SC) inoculated with PBS (Group A, animals #1, 2, 3) and intranasally challenged at 0 dpi with EBOV Kik-9510621 (Figure 1). Animals #1 and #2 were euthanized at 5 dpi, and pig #3 was euthanized at 6 dpi. Prior to the EBOV inoculation, animals exhibited normal behavior and the body temperature was within the normal physiological limit.

All results are summarized in Table 1 and Figure 2 and Figure 3. At postmortem examination, prominent gross pathology indicative of moderate to severe pneumonia, including hemorrhages and consolidation of lungs typical of EBOV infection, was observed in all the EBOV-challenged PBS control animals. Histopathology analysis confirmed that all pigs in this group developed prominent lesions indicative of moderate to severe pneumonia associated with hemorrhages and massive cell infiltration (Figure 2(Di) and Figure 3(Di)). Bronchial lymph nodes were enlarged and hemorrhagic, and the submandibular lymph nodes of two out of three pigs were enlarged as well. Broncho-alveolar lavage fluid (BALF) from all pigs was dark red in color and cloudy. One pig (animal #1) had petechial hemorrhages on the wall of the abdominal cavity and about 100 mL of very green stomach contents. Viral antigen was detected by immunostaining in lungs (Figure 2(Diii) and Figure 3(Diii)), both bronchial (Figure 2(Dv) and Figure 3(Dv)) and submandibular lymph nodes, as well as in the nasal turbinates (not shown) of all three animals in the control group.

Overall, the findings for the PBS-treated group challenged with infectious EBOV were consistent with our previously published data demonstrating that pigs can be infected experimentally displaying clear signs of EBOD [10,11].

### 2.2. Treatment with Ad5-porIFNλ3 Did Not Protect Pigs against EBOV

In parallel to the treatment with PBS, we evaluated the efficacy of treatment with Ad5-poIFNλ3 against EBOV. Prior to use, the expression of porIFNλ3 protein from the Ad5 vector was verified by western blotting and by standard measurement of bioactivity against VSV in MDBK cells. As previously reported, multiple protein bands were detected in cell lysates and supernatants of Ad5-porIFNλ3-infected HEK 293 cells, consistent with a pattern of glycosylation (Figure 2A) and cell supernatants displayed approximately 3000 U/mL of antiviral activity against reference VSV in MDBK cells [16]. It is worth mentioning that in the western blot (WB), a cross-reacting band of slightly higher MW as compared to PorIFNλ3, was also detected in Ad5 vector (namely Ad5-blue) infected cell extracts. Detection of this band was attributable to the use of Ad5-porIFNλ3 as the source of the antigen to derive the rabbit polyclonal antibody prepared in house at PIADC [15]. Four animals (group B) were SC inoculated with 2 × 10^10^ pfu of Ad5-porIFNλ3 (−1 dpi) followed by intranasal challenge with EBOV at 0 dpi. Similar to the animals in the control group (group A), two of the Ad5-porIFNλ3 treated pigs were euthanized at 5 dpi, and the other two pigs were euthanized at 6 dpi.

Although no clinical signs or behavioral changes were observed in any pig after the initial Ad5-porIFNλ3 administration, following the EBOV challenge, all pigs inoculated with Ad5-porIFNλ3 were mildly to moderately depressed and showed abdominal breathing. High levels of EBOV RNA were detected in nasal washes of all animals and in oral swabs of two of the Ad5-porIFNλ3-treated pigs (Figure 2B). The virus was isolated from the nasal wash of one Ad5-porIFNλ3-treated animal at 5 dpi (40 pfu/mL) (Figure 2C). After euthanasia, all pigs showed prominent gross pathology indicative of moderate to severe pneumonia, including hemorrhages and consolidation of lungs typical of EBOV infection (Figure 2(Dii)), similar to the PBS-treated animals (Figure 2(Di)). The lung lesions were associated with the presence of the viral antigen (Figure 2(Div)), as were the lesions in lymph nodes (Figure 2(Dvi)). Staining for the viral antigen in the tissues was comparable to staining in tissues of EBOV-challenged PBS control animals (Figure 2(Diii,Dv)). Viral RNA was consistently detected in the lungs, lymph nodes, liver, nasal turbinates, BALF, and BALP of all animals (Figure 2E), confirmed by virus isolation in all animals in most of the analyzed tissues (Figure 2F).

These results (also summarized in Table 1) indicated that treatment with Ad5-porIFNλ3 did not protect animals from EBOV infection, and the observed pathology was practically identical to that observed in PBS-treated challenge control animals.

### 2.3. Administration of Ad5-porIFNα Attenuated EBOD and Prevented Virus RNA Shedding in Pigs

We then performed an animal experiment to evaluate the efficacy of Ad5-porIFNα against EBOV. Expression of the porIFNα protein from this vector was also confirmed by western blotting using cell supernatants and lysates of porcine IBRS2 cells infected with Ad5-porIFNα. As previously reported (14), single or multiple bands characteristic of glycosylated IFNα, were detected in the blot depending on the antibody used in the assay (Figure 3A). Ad5-porIFNα-induced antiviral activity was evaluated against reference VSV in porcine IBRS2 cells using the same cell supernatants. Consistent with previous studies, high values of anti-VSV activity (3.16 × 10^6^ U/mL) were elicited by the Ad5-porIFNα vector, and a single or multiband pattern was detected in the western blot depending on the mono- or polyclonal antibody used for detection of the porIFNα protein [19]. Four pigs (group C) were SC inoculated with Ad5-porIFNα at −1 dpi and challenged intranasally with EBOV (0 dpi), followed by evaluation of clinical signs and pathology postmortem. None of the animals inoculated with the Ad5-porIFNα (group C) showed side effects after vector inoculation. Interestingly, none of the other animals in this group developed measurable clinical signs after the EBOV challenge except for one animal (pig#7) that appeared mildly depressed at 5 dpi (Table 1). Similarly, no viral RNA (Figure 3B) or infectious particles (Figure 3C) were detected in any nasal washes or oral swabs from Ad5-porIFNα pre-treated pigs. Analysis of the postmortem collected tissues indicated that no infectious virus was detected in three animals of the group; however, one animal (#10) had viral RNA in BALF, BALP, and bronchial lymph node (BLN) and infectious virus in BALF, albeit at lower amounts compared to the PBS-treated control animals (Figure 3E,F).

There was a striking difference in histopathology and virus antigen load between tissues collected from the Ad5-porIFNα-treated pigs compared to the PBS control animals (Figure 3D). Microscopically, significant lung lesions were detected in the control group A (Figure 3(Di)) correlating with the detection of viral antigen in two out of the three animals (Figure 3(Diii)), lesions in pig #1). In turn, viral antigen was detected in bronchial lymph nodes of all PBS-treated animals (Figure 3(Dv)). Lung lesions were characterized by edema, fibrin, and degenerating neutrophils within alveolar spaces. Alveolar walls were expanded by inflammatory cells (alveolitis), and there was perivascular and peribronchiolar cuffing presumably caused by the presence of mononuclear inflammatory cells. There was also evidence of type II pneumocyte hyperplasia and bronchiolitis. In contrast, no significant microscopic lesions or viral antigens were detected in the lungs (Figure 3(Dii–Div)), or in bronchial lymph nodes (Figure 3(Dvi)) of any of the animals treated with Ad5-porIFNα (group C).

These results clearly indicated that treatment with Ad5-porIFNα dramatically reduced virus replication and protected pigs against EBOV pathogenesis, completely abolishing virus shedding.

### 2.4. Analysis of the Response to Ad5-porIFNα Treatment

In order to better understand the mechanisms of protection elicited by Ad5-porIFNα the systemic response to the inoculation with this vector was evaluated by measuring the amount of IFNα protein and its biological activity in plasma, and by performing transcriptome (mRNA expression) analysis in PBMCs. The same analysis was not performed in the Ad5-porIFNλ3-treated group, given the observed lack of protection against EBOD, and previous data demonstrating that type III IFN was not active in peripheral blood cells [20]. Further, no systemic antiviral activity was detected using the same batch of vector in an efficacy experiment performed in swine against challenge with FMDV, albeit full protection from clinical disease had been observed [15]. Similarly, no protection or significant changes in IFN and ISG mRNA expression or systemic antiviral activity were detected in animals inoculated with the empty vector Ad5-blue (control) at 1 day prior to challenge with FMDV [15].

The levels of systemic porIFNα were determined with an in-house developed sandwich ELISA. In Ad5-porIFNα-treated animals, IFNα protein rose above baseline, reaching a maximum (2 to 3-fold) at 24 h post treatment (0 dpi) and returned to baseline by 3–6 dpi (Figure 4A).

In contrast, the levels of porIFNα in PBS control pigs did not change throughout the experiment despite the detection of viremia (Figure 4A).

To further assess the biological impact of the porIFNα protein expressed from the Ad5-porIFNα vector prior to EBOV inoculation, microarray analysis was performed on RNA extracted from PBMCs at 24 h post Ad5-vector inoculation (0 dpi) using RNA extracted from PBMCs isolated from animals treated with PBS as reference. Experimental microarray data were submitted to the Interferome database (www.interferome.org, accessed on 26 July 2019). As seen in Figure 4B, 522 genes were upregulated, and 171 genes were downregulated (2-fold or more) in the pigs treated with Ad5-porIFNα as compared to those treated with PBS (control animals). By using the DAVID bioinformatics suite, we identified KEGG pathways (Table 2) and GO biological processes (Table 3) represented in the upregulated dataset. As expected, these analyses primarily identified a set of genes involved in immune responses, including those encoding for chemokine-mediated signaling proteins, proteins involved in inflammation, proteins involved in immune cell chemotaxis, and other immune mediators. Another interesting category consisted of a number of genes involved in the biosynthesis of antibiotics, including a collection of metabolic enzymes (Table 2).

About fifty percent of altered genes (283 porcine genes) were characterized as induced by type I/II IFN, 25% (143 porcine genes) were characterized as type I IFN inducible, 25% (142 porcine genes) as type II IFN inducible, and a single gene was identified as type III IFN inducible.

A complete list of the results has been deposited in *NCBI GEO* GSE162846 (https://www.ncbi.nlm.nih.gov/geo/query/acc.cgi?acc=GSE162846, accessed on 9 December 2020).

The largest category of induced genes included cytokines and chemokines that affect T-cell differentiation (IL27) and T-cell recruitment (CCL5, CCL23, CCL24, CXCL9, CXCL11). A variety of receptors with immune functions were also upregulated (CXCR4, ACKR1, LTB4R, CD274, SYK); several cytokines that induced proliferation and activation of natural killer cells (CCL5 and IL15) were upregulated, as well as one protein involved in the attraction of lymphocytes (CXCL1). In addition, the mRNA of genes involved in various other immune responses and regulatory functions (TNFS13B, TNFAIP3, NFIL3, and OAS2) were upregulated (Figure A1).

Altogether, these results confirmed that Ad5-porIFNα expressed a bioactive molecule at the time of inoculation with EBOV.

## 3. Discussion

In the present study, we demonstrated that prophylactic treatment with Ad5-porIFNα is highly effective at protecting pigs from infection and virus shedding after exposure to EBOV. Treatment with Ad5-porIFNα completely prevented virus shedding, and not even viral RNA could be detected in nasal or oral swabs. Highly reduced virus replication in tissues, only detectable as RNA in a few samples, strongly supported this finding. Attempts for virus isolation were only successful in one BALP pellet but at low virus titers.

This outcome correlated well with previous studies in which prophylactic treatment with Ad5-porIFNα attenuated or fully prevented other viral infections of livestock animals, such as those caused by FMDV [21] or Classical Swine Fever Virus (CSFV) [15].

Interestingly, here we also demonstrated that the inhibitory effect was type I IFN (IFNα) specific, since treatment with an identical Ad5 vector expressing another type of IFN, porcine type III IFN (IFNλ3) did not protect this species against EBOV infection. This lack of protection by IFNλ3 was disease specific since we had previously demonstrated that similar doses of Ad5-porIFNλ3 inoculated in swine was indeed effective against FMD [15]. Selective antiviral activity may depend on the disease pathogenesis, most likely the type of cells infected by the virus, and their ability to respond to different types of IFN. In this regard, it is known that IFNλ3 preferentially induces antiviral responses in cells of epithelial origin, usually those present in skin and mucosa [19]. In contrast, EBOV targets specific cell types, such as liver cells, cells of the immune system, and endothelial cells [22], resulting in a very different disease pathogenesis, as compared to FMDV, which primarily replicates in epithelial cells [23].

In trying to understand the mechanism by which porIFNα altered EBOV-induced pathways, we performed microarray analysis to evaluate the response of EBOV-infected pigs that were pre-treated with Ad5-porIFNα in comparison to the PBS-treated control animals. Our data expanded the analysis of porcine genes and pathways that were altered in response to porIFNα treatment in vivo and overlapped 100% with the human and mouse interferome (www.interferome.org, accessed on 26 July 2019). In this regard, we deposited our data in the same database for public access. It is worth mentioning that in previous experiments in pigs, we had evaluated the innate response induced by inoculation with Ad5-blue, the empty parental vector of Ad5-porIFNα, demonstrating that no significant changes in a battery of ISGs mRNAs could be detected at 6 or 24 h post treatment [15,24]. Therefore, considering the limitations inherent to work in a BSL4 environment, and animal welfare ethics, an extra group of animals with this Ad5-blue control was not considered strictly necessary for this work.

In this study, at 3 days post EBOV infection, pigs that had been pre-treated with Ad5-porIFNα displayed upregulation of many genes related to the immune response in contrast to pigs treated with PBS. The data from our current microarray could be somewhat compared to a previous microarray in which we characterized the expression of selected genes in lung tissue from EBOV-infected and uninfected pigs [25]. Interestingly, only two genes (OAS2 and IL27) were identified in both studies as upregulated. This may suggest that pre-treating pigs with IFNα prior to EBOV exposure suppressed the cytokine storm caused immunopathology observed in the EBOV-infected pigs [25], heightened immunity against this viral infection, and substantially altered the nature of the immune response mounted by EBOV exposure, for example, by increasing T-and natural killer- cell responses. Interestingly, both natural killer [26] and dendritic cell [27] responses are known to be protective against EBOV in mouse and NHP models. Identifying the target cells of EBOV infection in pigs will be crucial in understanding differences in viremia and pathogenesis. Furthermore, these studies may help to understand the differential response observed between treatment with IFNα vs IFNλ3, both of which transduce similar signals but in a tissue-specific manner depending on the selective expression of the type III IFN receptor [20].

Other studies have previously measured PBMC transcriptional responses in human and NHP Ebola cases. In all of these studies, ISGs, cytokines, chemokines, T cell signaling, and other immune pathways could also be found prominently upregulated at the mRNA level, despite the presence of severe disease [28]. Generally, pro-inflammatory cytokines and chemokines and markers of inflammation are associated with poor disease outcome or death, as are markers of endothelial function and coagulation. In contrast, recovery and survival, have been associated with higher levels of gastric tissue integrity markers and T-cell responses [29]. However, it is currently unknown whether transcriptomic responses to EBOV are recapitulated at the protein level, and it is difficult to correlate this knowledge of human EBOV disease to disease caused by EBOV in pigs, as they are rather different: very little viremia and respiratory tract are the most significant physiological systems involved in swine compared to high viremia and involvement of multiple systems in human disease.

Our current study, interestingly, also identified several other biological processes as being differentially regulated in PBMCs of Ad5-porIFNα pre-treated pigs as compared to control PBS-treated pigs after challenge with EBOV. For example, we identified the calcium signaling pathway, which, to our knowledge, has not been previously associated with IFN signaling or antiviral mechanisms. Consistently, a recent study reported that a calcium channel blocker had antiviral activity against EBOV and the Marburg virus in vivo [30].

Although in these experiments, we did not perform the microarray analysis in the animals treated with Ad5-porIFNλ3, it will be interesting to determine if there are differential gene expression patterns in these animals in comparison to those treated with Ad5-porIFNα. However, rather than studying PBMCs, analysis of tissue resident immune cells would be desirable since PBMCs do not respond to IFNλ. As mentioned above, the expression of type III IFN receptor is almost exclusive to mucosal cells, and therefore significant changes are only expected in mucosal tissues [20]. Although the EBOV challenge involved intranasal deposition of the virus, a broader cell-specific virus tropism or saturation of the system by the virus infection may explain the lack of a protective response by the treatment with Ad5-porIFNλ3, even when it relied on similar patterns of gene induction. A higher dose of Ad5-porIFNλ3 or an alternative route of delivery (i.e., intraoral/intranasal) may be required to limit EBOV replication by this IFN in these tissues.

Overall, this proof-of-concept study clearly demonstrates the efficacy of Ad5-porIFNα pre-treatment against EBOV in pigs and highlights the potential of its use for outbreak control. Nevertheless, using this vector as a control strategy during an outbreak presents challenges. For example, during livestock outbreaks, the first intervention is typically depopulation of the indexed farm and implementation of ring vaccination in surrounding farms. Treating animals with porIFNα could allow to slow down virus spread, while immunity induced by vaccination is established. In fact, if the Ad5-porIFNα vector were to be used in the field, the timing and the window of effectiveness would be critical, and different formulations to prolong its expression or bioavailability may need to be developed, as well as exploration of more economic substitutes. The use of recombinant IFN protein produced in *E. coli* or in baculovirus seems an attractive alternative strategy. In this regard, it has been shown that the half-life of IFN could be extended by chemical modification or fusion to other proteins. Remarkably, we have recently demonstrated that pegylated porIFNα could effectively protect pigs against FMD even at 5 days post inoculation [31]. Ultimately, an IFN based therapy that induces early innate immunity, in combination with an effective vaccine that could induce fast-adaptive immunity in one week or less, may provide a solution to fully protect animals against EBOV in case of an outbreak. In proof-of-concept studies, this combination approach has been effective in providing full protection of pigs for an animal disease of outmost importance such as FMD [14]. Such an approach would ultimately protect pigs, but most importantly, it could prevent EBOV transmission to humans.

Thus, a fine tuning of IFN administration and formulation, timing, dose, route of inoculation, and window of effectiveness warrants further investigation to develop tools that could effectively control and/or prevent emergent diseases of high importance, such as EBOD.

## 4. Materials and Methods

### 4.1. Cell Lines and Viruses

The 1995 Kikwit 9510621 isolate of EBOV was kindly provided by Dr. Kobinger, NML, PHAC. The stock virus was produced and titrated on Vero E6 (ATCC CRL-1586) cells in minimal essential medium (MEM) supplemented with 2% fetal bovine serum (FBS) and antibiotics. EBOV titers were determined by a microtiter plaque assay, as previously described [32]. Porcine kidney IBRS2 cells were obtained from USDA-APHIS at PIADC and maintained in MEM supplemented with 10% FBS supplemented with nonessential amino acids and antibiotics. The VSV serotype New Jersey (VSNJV) field strain (95COB) was obtained from USDA-APHIS at PIADC. Human 293 cells (HEK 293, ATCC CRL-1573) and Madin-Darby bovine kidneys (MDBK, ATCC CCL-22) were purchased from ATCC. HEK293, MDBK, and IBRS2 cell lines were maintained in MEM supplemented with 10% FBS, non-essential amino acids, and antibiotics.

### 4.2. Design and Construction of the Ad5-porIFN Vectors

Replication-defective human Adenovirus 5 (E1 and E3 deleted) expressing porcine IFNλ3 or por-IFNα were constructed by cloning the sequence of the IFNλ3 gene (IL28B, Gen Bank FJ853389, [15]) or the IFNα gene 1 (GenBank X57091 [20] into the E1 deleted region of pAd5-blue [32]. Recombinant pAd5-porIFNλ3 or pAd5-porIFNα were linearized with *Pac*I and transfected into HEK 293 cells using Lipofectamine^R^ following the manufacturer’s directions (Thermo Fisher, Waltham, MA). Recovered Ad5-porIFNs were amplified in HEK293 cells and purified by cesium chloride gradient density centrifugation. The development and production of the vector master seed was done at the Plum Island Animal Disease Center (PIADC) and sent to Vector Biolabs (Malvern, PA, USA) for propagation. High titer stocks were shipped to the National Center for Foreign Animal Disease (NCFAD), Winnipeg, Manitoba, Canada. Titers of the recombinant Ad5 viruses were determined by end point dilution as previously described [14] using HEK 293 cells in MEM supplemented with 2% fetal bovine serum, and antibiotics. Quality control assays of the specific Ad5-porIFN batches used in the experiments were performed by western blot analysis and measurement of bioactivity against VSV as previously described [15]. PorIFNλ3 was detected using a rabbit polyclonal antibody made at PIADC [15]. PorIFNα was detected with mAb K18 (PBL Cat # 27100-1) or a rabbit polyclonal antibody against porIFNα made at PIADC.

### 4.3. Animal Study

All procedures involving infectious Ebola virus (EBOV), including pig inoculation, were performed in the biosafety level 4 (BSL4) containment facility at the NCFAD in Winnipeg, Canada. Five-week-old pigs (Landrace x Large White x Duroc cross), weighing approximately 15 kg each, were obtained from a recognized commercial supplier in Manitoba, Canada, and acclimated in the BSL4 animal facility for one week. All inoculations and samplings were done under inhalation anesthesia. Clinical signs, behavioral changes, and rectal temperature were recorded daily. Group A (animals #1, 2, 3) were SC inoculated in the neck with PBS; Group B (animals # 4, 5, 6, 7) was SC inoculated in the neck with 2 × 10^10^ pfu/2mL of Ad5-porIFNλ3; Group C (animals # 8, 9, 10, 11) was SC inoculated in the neck with 2 × 10^10^ pfu/2mL of Ad5-porIFNα. Due to space constrictions in BSL4, the study was performed consecutively, inoculating pigs with PBS, Ad5-porIFNλ3 or Ad5-porIFNα. On the first day of the experiment (−1 dpi), three control pigs (Group A: Pig # 1, 2, 3) were SC in the neck with 2 mL of PBS, or with 2 × 10^10^ pfu Ad5-porIFNλ3/2mL PBS (Group B: # 4, 5, 6, 7) or 2 × 10^10^ pfu Ad5-porIFNα/2mL PBS (Group C: Pig # 8, 9, 10, 11). Twenty-four hours later (0 dpi), all pigs were oro-nasally challenged with 10^6^ pfu EBOV/2mL PBS per pig (0.5 mL per each nostril, and 1 mL orally). Nasal washes, oral swabs, and blood were collected for each pig at −1 dpi, and at 0, 3, and 5 or 6 dpi. An overview of the inoculation and sampling is provided in Figure 1. Blood was collected into BD vacutainer CPT^TM^ cell preparation tubes (Becton Dickinson, New Jersey, NJ, USA), plasma, and peripheral blood mononuclear cells (PBMC) were separated according to the manufacturer’s protocol. PBMC pellets diluted four times with PBS, oral swabs, nasal washes, and plasma were stored at −80 °C prior to analysis. Two PBS control, two Ad5-porIFN λ3 and two Ad5-porIFNα inoculated pigs were euthanized at 5 dpi; all the remaining pigs were euthanized at 6 dpi. Postmortem findings were recorded, and collected tissues were either fixed in paraformaldehyde for analysis by histopathology or stored at −80 °C for virus and RNA analysis, as were the broncho-alveolar fluid (BALF) and cell pellet (BALP) samples.

### 4.4. EBOV Detection

Nasal washes, oral swabs, plasma, tissues, PBMCs, BALF, and BALP were collected as outlined in Figure A1. Tissue was homogenized in 5 mL PBS (10% *w*/*v* homogenates) using a lysing kit and homogenizer from Precellys^®^ according to the manufacturer’s protocol (BERTIN Corp., Rockville, MD, USA). Total RNA was extracted from 100 µL of liquid samples in 900 µL TriPure (Roche) and from the whole PBMC and BALP pellets lysed in 1 mL TriPure according to the manufacturer’s protocol. The amount of EBOV in each sample was determined by a semi-quantitative real-time polymerase chain reaction (qRT-PCR) assay targeting the L-gene [31]. The detection limit for this test was 3.6 log_10_ RNA copies/mL. Samples that were PCR-positive were subjected to virus isolation using a microtiter immunostained plaque assay as previously described [31].

### 4.5. Histopathology and Immunohistochemistry

Tissues were fixed in 10% neutral phosphate buffered formalin, embedded in paraffin using standard procedures, sectioned at 5 µm, and stained with hematoxylin and eosin (HE) using standard procedures. Detection of the EBOV VP40 antigen by immunohistochemistry was done on serial sections matching the HE-stained tissue sections as previously described [10].

### 4.6. Porcine IFNα Detection in Plasma by ELISA

Ninety-six-well microtiter plates (NUNC Maxisorb, Thermo Fisher Scientific Inc., Waltham, MA, USA) were coated with 50 μL/well of anti-pig IFNα antibody, Clone K9 (1.66 µg/mL, PBL Assay Science) at a dilution of 1:1000 in carbonate–bicarbonate buffer (Sigma-Aldrich Inc.), and incubated overnight at 4 °C. The plates were then washed five times with PBST (PBS with 0.05% Tween 20, pH 7.2), followed by the addition of 50 µL plasma or positive control recombinant porIFNα (Kingfisher Biotech, St. Paul, MN, USA) and incubated 1 h at 37 °C while shaking, and then washed 5 times with PBST. Fifty microliters of 1:1000 biotinylated anti-pig IFN-α antibodies (Clone F17, 1.66 µg/mL, PBL Assay Science) were added to the plate followed by 1 h incubation at 37 °C with shaking. The plates were washed as described above, and 50 µL 1:1000 ExtrAvidin^®^ (BTAG, Sigma-Aldrich Inc., Burlington, MA, USA) was added and incubated for 30 min at 37 °C while shaking. Plates were again washed and developed by adding 100 µL of tetramethylbenzidine solution (TMB, # 34028, Thermo Scientific, Waltham, MA, USA), followed by 15 min incubation in the dark. The color development was stopped by adding 50 µL of 2M H_2_SO_4_ and the optical density (OD) was measured at 450° nm using a spectrophotometer (SpectraMax Plus 384 Microplate Reader, Molecular Devices, San Jose, CA, USA). Concentrations of plasma porIFNα were calculated based on the standard curve generated by serial dilutions of recombinant porcine IFNα (Kingfisher Biotech, Inc., St Paul, MN, USA) and expressed as nanograms per mL (ng/mL).

### 4.7. Microarray & Bioinformatics

Microarray analysis was performed in samples obtained from Ad5-porIFNα- and PBS-treated animals at 1 day post treatment, prior to challenge with EBOV. Total RNA was extracted from PBMC and BALP, and the concentrations were determined using a NanoDrop (NanoDrop Technologies) and shipped to MOGENE LC (St. Louis, MO, USA) for the microarray assay. Microarray analysis was performed using an Agilent porcine 4x44K microarray (Agilent Technologies) and following the manufacturer’s protocols. Microarray slides were scanned on an Agilent high resolution C scanner and images processed using the Agilent Feature Extraction software. The quality of samples was analyzed using the ArrayQualityMetrics package and the data imported into GeneSpring 14.8 (Agilent) for further analysis. The data was normalized using a shift to the 75th percentile; thereafter, the value was further transformed as log2. All gene IDs from genes that were up- or down-regulated > 2-fold were then imported into the DAVID bioinformatics suite (https://david.ncifcrf.gov/, accessed on 26 July 2019) for detecting overrepresentation of signaling pathways from the Kyoto Encyclopedia of Genes and Genomes (KEGG) [33] and gene ontologies (GO) [34].

FDR-values and p-values were calculated by the DAVID bioinformatics suite using an adapted method for multiple testing with independent statistics [35].

## Figures and Tables

**Figure 1 pathogens-11-00449-f001:**
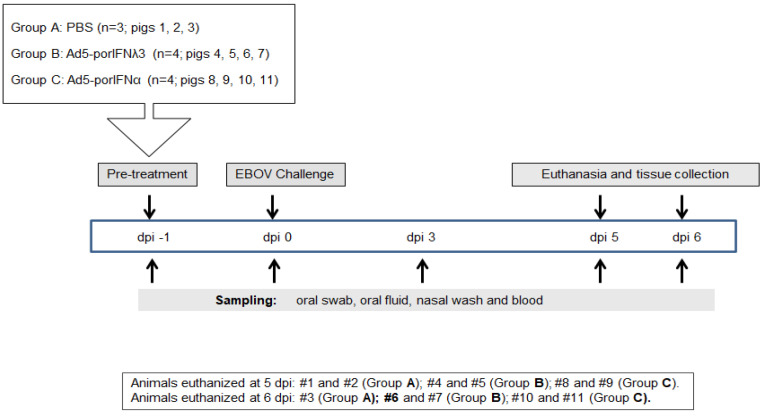
Schematic representation of the treatments, sampling, and euthanasia for all three groups. Animals were subcutaneously (SC) inoculated in the neck with PBS (Group A), Ad5porIFNλ3 (Group B) or Ad5porIFNα (Group C) one day prior (−1 dpi) to challenge with EBOV/Kik-9510621 (0 dpi) administered oro-nasally. Oral swab, oral fluid (rope chews), nasal washes, and blood were collected at −1, 0, 3, 5, and 6 dpi. Pigs were euthanized on either 5 or 6 dpi. PBMC: peripheral blood mononuclear cells.

**Figure 2 pathogens-11-00449-f002:**
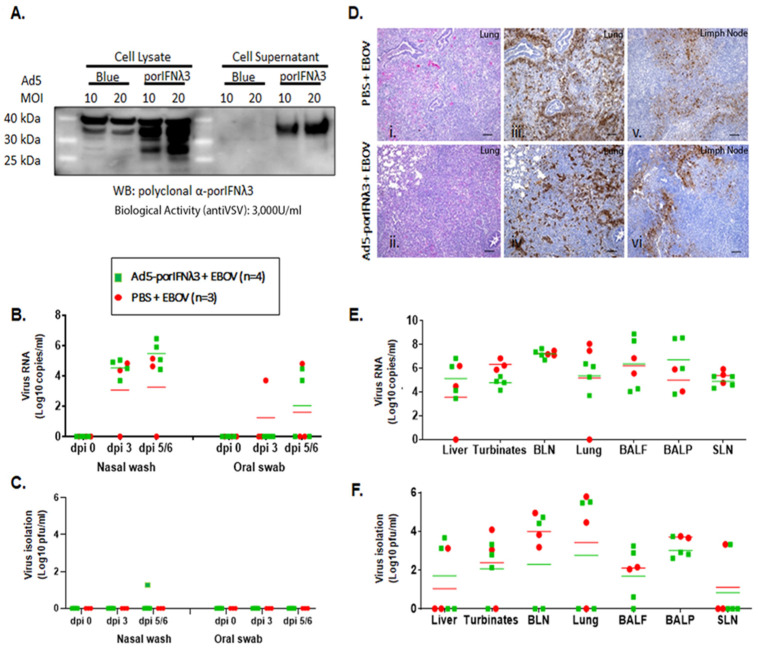
Ad5-porIFNλ3 administration did not protect against EBOV infection. (**A**): Ad5 expression of porIFNλ3 was determined by western blot (WB) analysis using a rabbit polyclonal antibody made at PIADC. Multiple specific bands represent different glycosylation stages of the IFNλ3 protein. Biological activity was determined in vitro in MDBK cells using VSV as a reference virus. (**B**): Detection of viral RNA by real-time RT-PCR targeting the EBOV L gene in nasal washes or oral swabs. (**C**)**:** Detection of virus as plaque forming units (pfu) in nasal washes or oral swabs; plaques were visualized by immunostaining with anti-EBOV VP40 (PBS—red circles; Ad5porIFNλ3—green squares). (**D**): Histopathology, hematoxylin/eosin (HE) and immunostaining in tissues. (**i**,**ii**) Lungs (Group A, PBS; Group B, Ad5porIFNλ3). Both groups developed severe broncho-interstitial pneumonia by 5 dpi with EBOV. (**iii**,**iv**) Detection of EBOV by immunostaining with anti-EBOV VP40 in lungs (Group A, PBS; Group B, Ad5porIFNλ3): (**v**–**vi**) Detection of EBOV by immunostaining with anti-EBOV VP40 in lymph nodes (Group A, PBS; Group B, Ad5porIFNλ3). Horizontal bar = 100 µm. (**E**): Detection of EBOV RNA (real-time RT-PCR) in 10% *w*/*v* tissue homogenates. (**F**): Detection of EBOV by virus isolation in tissue homogenates; plaques were visualized by immunostaining with anti-EBOV VP40 protein. (PBS—red circles; Ad5porIFNλ3—green squares).; BLN, bronchial lymph node; SLN, sub-mandibular lymph node; BALF, broncho-alveolar lavage fluid; BALP, broncho-alveolar lavage pellet.

**Figure 3 pathogens-11-00449-f003:**
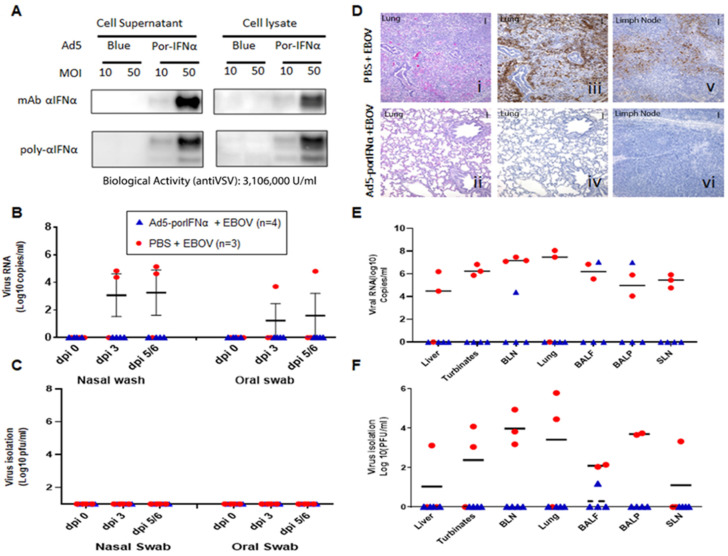
Ad5-porIFNα administration protected against EBOV infection. (**A**): Ad5 expression of porIFNα was determined by western blot analysis using a polyclonal antibody made at PIADC. Multiple bands represent different glycosylation stages of the IFN protein. Biological activity was determined in vitro in IBRS2 cells using VSV as a reference virus. (**B**): Detection of viral RNA by real-time RT-PCR targeting the EBOV L gene in nasal washes or oral swabs. (**C**): Detection of virus as plaque forming units (pfu) in nasal washes or oral swabs; plaques were visualized by immunostaining with anti-EBOV VP40 (PBS—red circles; Ad5porIFNα—blue triangles). (**D**): Histopathology, hematoxylin/eosin (HE) and immunostaining in tissues. (**i**,**ii**): Lungs (Group A, PBS; Group C, Ad5porIFNα). Severe broncho-interstitial pneumonia was observed in the lungs of animals treated with PBS by 5 dpi with EBOV. No detectable pathology was detected in the lungs of animals treated with Ad5porIFNα. (**iii**–**iv**): Detection of EBOV by immunostaining with anti-EBOV VP40 in lungs (Group A, PBS; Group B, Ad5porIFNα): (**v**–**vi**): Detection of EBOV by immunostaining with anti-EBOV VP40 in lymph nodes (Group A, PBS; Group B, Ad5porIFNα). No EBOV was detected by direct immunostaining of lungs or lymph nodes tissue sections derived from Ad5porIFNα treated animals. Vertical bar = 100 µm. (**E**): Detection of EBOV RNA (real-time RT-PCR) in 10% *w*/*v* tissue homogenates. (**F**): Detection of EBOV by virus isolation in tissue homogenates; plaques were visualized by immunostaining with anti-EBOV VP40 protein. (PBS—red circles; Ad5porIFNα—blue triangles).; BLN, bronchial lymph node; SLN, sub-mandibular lymph node; BALF, broncho-alveolar lavage fluid; BALP, broncho-alveolar lavage pellet.

**Figure 4 pathogens-11-00449-f004:**
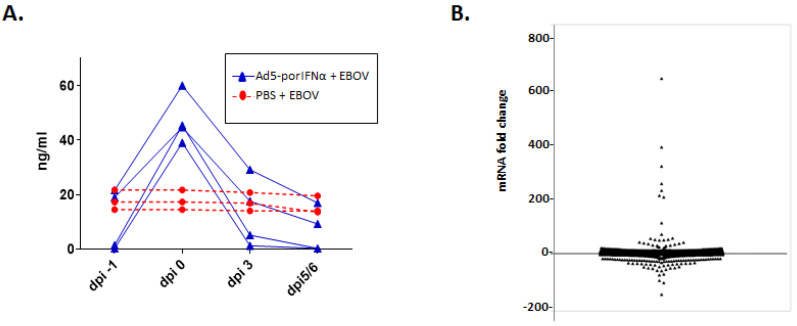
Treatment with Ad-5 porIFNα-treated induces elevated levels of systemic IFNα triggering the expression of multiple genes in peripheral blood mononuclear cells (PBMC). (**A**): Plasma levels of porIFNα were measured in animals of Groups A and C using a sandwich ELISA. IFNα protein concentrations were calculated from a standard curve generated with commercially available recombinant porIFNα. Dpi—days post-inoculation. Time 0 is indicated as day of exposure to EBOV, at 24 h post treatment with PBS or Ad5-porIFNα (group A—red circles; Group C—blue triangles). (**B**): Overall expression (up- and downregulation) of genes in PBMCs isolated at 24 h post treatment with Ad5-porIFNα relative to treatment with PBS.

**Table 1 pathogens-11-00449-t001:** Summary of the clinical signs, pathology, and detection of EBOV (virus or RNA).

	PM Testing (dpi)	Clinical Signs	Lung Pathology	Shedding (VI Nasal Wash)	Viral RNA (in Tissues)	Virus Isolation (in Tissues)
**Group A: PBS and EBOV**						
1	5	Depressed Increased RR	Pneumonia	+	+	+
2	5	Depressed	Pneumonia	+	+	+
3	6	Depressed	Pneumonia	+	+	+
**Group B: Ad5-porIFNλ3 and EBOV**						
4	5	No	Pneumonia	+	+	+
5	5	No	Pneumonia	+	+	+
6	6	No	Pneumonia	+	+	+
7	6	No	Pneumonia	+	+	+
**Group C: Ad5-porIFNα and EBOV**						
8	5	Mildly depressed	Normal	−	−	−
9	5	No	Normal	−	−	−
10	6	No	Normal	−	−	−
11	6	No	Normal	−	+	+
					(BLN, BALF, BALP)	(BLN, BALF, BALP)

PM, postmortem; Dpi, days post inoculation; VI, virus isolation; BALF, broncho-alveolar lavage fluid; BALP, broncho-alveolar lavage pellet; BLN, bronchial lymph node; RR, respiratory rate. “**+**” indicates “detected”; “**−**” indicates “absent”.

**Table 2 pathogens-11-00449-t002:** Top 10 upregulated KEGG pathways of IFN-induced genes in PBMCs. A list of all upregulated proteins was entered into the DAVID^®^ bioinformatic pipeline and submitted for KEGG^®^ pathway analysis. All pathways that were represented by the dataset were reported with a corresponding *p*-value (<0.05) and false discovery rate (FDR).

KEGG Category	Genes	*p*-Value	FDR
Endocytosis	RAB4A, LOC100621443, SMAD2, HSP70, ARF5, KIT, CAPZB, FOLR1, ARPC2, CXCR4, RAB35, GRK5, VPS26A	0.024	26.70
Cytokine-cytokine receptor interaction	AMHR2, CCL2, IL6ST, CXCL9, IL15, KIT, CXCL11, CCL5, IL7R, IL10, TNFSF13B, CXCR4, LTA, IFNGR1	0.019	21.68
Chemokine signaling pathway	CCL24, CCL2, CCL23, MAP2K1, CXCR4, PTK2B, CXCL9, GRK5, CXCL11, CCL5, XCL1, STAT3, CHUK	0.007	8.74
Herpes simplex infection	SRSF5, CCL2, TAF4B, HCFC2, ARNTL, OAS2, IL15, CCL5, PPP1CB, IFNGR1, CHUK, LTA	0.025	27.74
Biosynthesis of antibiotics	FNTB, PGP, ACADM, SQLE, OGDHL, BCKDHB, PLA2G7, PGAM2, PDHA1, CAT, OAT, HADHA	0.034	35.86
FoxO- signaling pathway	SGK1, MAP2K1, SMAD4/2, CAT, IL7R, INSR, ATM, CHUK, IL10, STAT3	0.008	9.67
cGMP-PKG- signaling pathway	EDNRA, MEF2C, FXYD2, MEF2A, MAP2K1, PPP3CB, GUCY1A3, PDE3A, INSR, PPP1CB, MYLK	0.034	35.52
Osteoclast Differentiation	MAP2K1/K6, FCGR2B, PPP3CB, TREM2, IFNGR1, CHUK, SYK, BTK	0.022	24.41
NF-kappa B- signaling pathway	ICAM1, TNFSF13B, TNFAIP3, ATM, CHUK, LTA, SYK, BTK	0.006	7.42
Carbon metabolism	PGP, ACADM, MCEE, OGDHL, PGAM2, PDHA1, CAT, HADHA, PC	0.014	16.55

**Table 3 pathogens-11-00449-t003:** Top 10 upregulated Gene Ontology Biological Processes of IFNα-induced genes in PBMCs. A list of all upregulated proteins was entered into the DAVID^®^ bioinformatic pipeline and submitted for gene ontology (GO) analysis. All biological processes that were represented by the dataset were reported with a corresponding *p*-value (<0.083) and false discovery rate (FDR).

KEGG Category	Genes	*p*-Value	FDR
Inflammatory response	LIPA, CCL2, IL27, CXCL9, ACKR1, KIT, CXCL11, CCL5, IL10, CALCB, CCL23, LTB4R, XCL1, SYK	0.003	5.40
Transcription, DNA-templated	MEF2C, MEF2A, LIN52, KLF9, ESR1, SMAD2, ARNTL, GTF2H2, STAT3, NCOA1, NCOA3, ZSCAN21, BHLHE41, RSC1A1	0.083	76.41
Immune response	CCL24, CCL23, TNFSF13B, CD274, CXCL9, OAS2, IL15, CXCL11, NFIL3, CCL5, TNFAIP3, XCL1, IL10	0.013	20.14
Cell adhesion	ICAM1, CD9, SIGLEC1, APP, PTK2B, ATP4B, TNC, RHOB, FES, ENG, SPP1	0.004	6.14
Chemokine-mediated signaling pathway	CCL2, CCL23, PTK2B, CXCR4, ACKR1, CXCL9, CCL5, CXCL11, XCL1	0.000	0.01
Upregulation of ERK1 and ERK2 cascade	ICAM1, ALOX15, CCL2, CCL23, PTK2B, ANGPT1, CCL5, XCL1, GAS6	0.012	17.74
Protein phosphorylation	APP, PHKB, PRKRA, PTPRA, PPP3CB, CDK4, MYLK, SYK	0.008	12.74
Positive regulation of gene expression	MEF2C, ACTA2, CD46, QKI, KIT, NFIL3, IL7R, GAS6	0.047	55.47
Regulation of cell proliferation	SGK1, PTK2B, TNC, CXCL9, EGLN3, JAG1, CXCL11, FES	0.060	64.01
Cell chemotaxis	CCL24, CCL23, HBEGF, KIT, CCL5, CXCL11, XCL1	0.004	6.90

## Data Availability

The datasets presented in this study can be found at NCBI GEO GSE162846.

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
