# Peer review of "Treatment with Ad5-Porcine Interferon-α Attenuates Ebolavirus Disease in Pigs"

_pathogens, 2022, doi:10.3390/pathogens11040449_

Round 1

Reviewer 1 Report

The study focused on the capacity of IFNa to protect pig from an EBOV challenge. The paper is well written and organized. Experimental section is clear and figures are cleared too. A large part of references is recent (less than five years old). Working with animal model in BSL4 is still a challenge then the number of pig in this study is limited to 3 to 4 animals per group. Some positive controls were performed to validate the pAd5-porIFNa capacity to reduce VSV and FMDV infectivity on a cell model. Experimental studies showed the capacity of IFNa to prevent severity of EBOD in pigs while no effect was obtained with type III IFN. Microarray analysis tried to identify some actors involved in the IFNa efficacy to reduce EBOD in pigs. This fastidious analysis was performed to investigate gene expression pattern when animals were treated with IFNa. Supplementary data are helpful for the reader.

The major limitation of the paper results in the kinetic of activity of aIFN to control or prevent EBOD because protection is effective in a very limited window of 24h before EBOV infection. It is true that the strategy to combine aIFN with a pegylated protein displayed a 5 days antiviral activity against FMDV (Diaz-San Segundo et al, 2021).  

Minor points:

Figure 1:

Panel 1A: can authors explain protein bands in cell lysate for the control condition (Blue) identified with the polyclonal porIFNl3?

Panel 1D: to help the reader, up to the panels can be indicated the type of tissue (lung, lung, lymph nodes)

Author Response

Reviewer 1

The study focused on the capacity of IFNa to protect pig from an EBOV challenge. The paper is well written and organized. Experimental section is clear and figures are cleared too. A large part of references is recent (less than five years old). Working with animal model in BSL4 is still a challenge then the number of pig in this study is limited to 3 to 4 animals per group. Some positive controls were performed to validate the pAd5-porIFNa capacity to reduce VSV and FMDV infectivity on a cell model. Experimental studies showed the capacity of IFNa to prevent severity of EBOD in pigs while no effect was obtained with type III IFN. Microarray analysis tried to identify some actors involved in the IFNa efficacy to reduce EBOD in pigs. This fastidious analysis was performed to investigate gene expression pattern when animals were treated with IFNa. Supplementary data are helpful for the reader.

The major limitation of the paper results in the kinetic of activity of aIFN to control or prevent EBOD because protection is effective in a very limited window of 24h before EBOV infection. It is true that the strategy to combine aIFN with a pegylated protein displayed a 5 days antiviral activity against FMDV (Diaz-San Segundo et al, 2021).  

We very much thank the reviewer for his/her positive comments in our manuscript. Below we have addressed his/her minor concerns.

Minor points:

Figure 1:

Panel 1A: can authors explain protein bands in cell lysate for the control condition (Blue) identified with the polyclonal porIFNl3?

We thank the reviewer for the observation. We have added two sentences in the results (lines 192-196 for IFN lambda3; and lanes 271 to 273 for IFNa) clarifying this issue

It is worthy to mention that in the western blot (WB), a cross reacting band of slightly higher MW as compared to PorIFNλ3, was also detected in Ad5 vector (namely Ad5-blue) infected cell extracts. Detection of this band was attributable to the use of Ad5-PorIFNλ3 as the source of antigen to derive the rabbit polyclonal antibody prepared in house at PIADC [15].

Consistently with previous studies, high values, 3.16x106 U/ml, of anti-VSV activity, were elicited by the Ad5-porIFNα vector, and a single or multiband pattern was detected in the western blot depending on the mono- or polyclonal antibody used for detection of the porIFNα protein [19]

Panel 1D: to help the reader, up to the panels can be indicated the type of tissue (lung, lung, lymph nodes)

We thank the reviewer for his/her suggestions and added the name of the tissue on each panel of the now named Figures 2 and 3.

Reviewer 2 Report

This is an important and well-performed study. The manuscript is well written, and the data are presented clearly. Nevertheless, the manuscript could be further improved by addressing the following minor points:

  1. LINE 51: Citation needed for a statement about RESTV asymptomatic in humans but lethal in NHPs.
  2. LINE 115: Some introduction to the use of Ad5 vectors for IFN delivery would be a useful addition to the Introduction.
  3. LINE 127: Strain/isolate should be indicated.
  4. LINE 189: Can the VSV antiviral cell culture data for both Ad5 vectors be included as a table/figure rather than simply a summary statement of 'approximate IU/mL?'
  5. LINE 189: Here and elsewhere in the Results indicate the route of inoculation.
  6. LINE 646: Experimental design figure should be included as a main figure rather than supplemental.
  7. Authors should mention whether levels of the vector-driven IFN RNA were detected by the microarray.
  8. Figure 3B a volcano plot including p-value/FDR and with top mRNA changes annotated would be more informative than the current plot. Also, consider this plot type for data in Supplementary Figure 1.
  9. Data appearing in Supplemental Figure 1 should be described in the Results section (instead of only the Discussion). Furthermore, text on heatmaps is too small/at too low of a resolution to make out.
  10. LINE 412: Here and elsewhere, the text should indicate that transcripts (mRNA) and not proteins are being measured.

Author Response

This is an important and well-performed study. The manuscript is well written, and the data are presented clearly. Nevertheless, the manuscript could be further improved by addressing the following minor points:

We very much thank the reviewer for his/her positive comments in our manuscript. Below we hope to have addressed his/her minor concerns.

  1. LINE 51: Citation needed for a statement about RESTV asymptomatic in humans but lethal in NHPs.

We thank the reviewer for his/her careful critical reading of our manuscript. We have added a reference with this information. Number [2] in our list.

P.B. Jahrling, T.W. Geisbert, E.D. Johnson, C.J. Peters, D.W. Dalgard, W.C. Hall,

  1. Preliminary report: isolation of Ebola virus from monkeys imported to USA,

The Lancet, 335, 502-505

 LINE 115: Some introduction to the use of Ad5 vectors for IFN delivery would be a useful addition to the Introduction.

We apologize for the omission. The text in lines 102-106 has been modified accordingly to include the use of Adenovirus vectors (replication defective human Adenovirus 5) in the introduction.

 LINE 127: Strain/isolate should be indicated.

We apologize for the omission of the strain ID in this section of the manuscript. It had only been included in M&M. We have now added EBOV/Kik-9510621 in line 128 and also in the description of the experimental scheme now included in the main manuscript.

  1. LINE 189: Can the VSV antiviral cell culture data for both Ad5 vectors be included as a table/figure rather than simply a summary statement of 'approximate IU/mL?'

As suggested by the reviewer the data of in vitro anti VSV antiviral activity induced by Ad5-IFN vectors has been also incorporated in panels A of both, Figures 2 and 3.

  1. LINE 189: Here and elsewhere in the Results indicate the route of inoculation.

We apologize for the omission. We have added the route of inoculation for PBS, the Ad5-IFNs and EBOV throughout the manuscript (subcutaneous for PBS and Ad5’s, and intranasal for EBOV).

  1. LINE 646: Experimental design figure should be included as a main figure rather than supplemental.

We agree with the reviewer and have included the experimental design in the main manuscript as Figure 1.

  1. Authors should mention whether levels of the vector-driven IFN RNA were detected by the microarray.

We thank the reviewer for his/her observation. We have added a sentence (lines 356-359) Similarly, no protection or significant changes in IFN and ISG mRNA expression or systemic antiviral activity had been detected in animals inoculated with the empty vector Ad5-blue (control) at 1 day prior to challenge with FMDV [15].

  1. Figure 3B a volcano plot including p-value/FDR and with top mRNA changes annotated would be more informative than the current plot. Also, consider this plot type for data in Supplementary Figure 1.

We agree with the reviewer about the information other type of graphs may provide, however by using our plots we are just trying to give a snapshot of the number of genes that were up- or downregulated by the Ad5-porIFNalpha treatment. Consistent information is depicted in the heatmaps of supplementary figure 1 -whose resolution has been improved-, and in the tables. In addition, all microarray data have been made public and is referenced in our manuscript

“complete list of the results has been deposited in NCBI GEO GSE162846 (https://www.ncbi.nlm.nih.gov/geo/query/acc.cgi?acc=GSE162846)”. WE hope the reviewer agrees with our assessment of the issue.

 Data appearing in Supplemental Figure 1 should be described in the Results section (instead of only the Discussion). Furthermore, text on heatmaps is too small/at too low of a resolution to make out.

We thank the reviewer for his suggestion and described the data of supplementary figure 1 in the results section of the manuscript (lines 415-431).

We have also improved the resolution of Supplementary Figure 1.

  LINE 412: Here and elsewhere, the text should indicate that transcripts (mRNA) and not proteins are being measured.

We thank the reviewer for the careful reading of our manuscript. We have replaced the word protein by mRNA or vice versa, wherever it was appropriate.